# Titanium Dioxide Thin Films Obtained by Atomic Layer Deposition Promotes Osteoblasts’ Viability and Differentiation Potential While Inhibiting Osteoclast Activity—Potential Application for Osteoporotic Bone Regeneration

**DOI:** 10.3390/ma13214817

**Published:** 2020-10-28

**Authors:** Agnieszka Smieszek, Aleksandra Seweryn, Klaudia Marcinkowska, Mateusz Sikora, Krystyna Lawniczak-Jablonska, Bartlomiej. S. Witkowski, Piotr Kuzmiuk, Marek Godlewski, Krzysztof Marycz

**Affiliations:** 1Department of Experimental Biology, Wroclaw University of Environmental and Life Sciences, Norwida St. 27 B, PL-50375 Wroclaw, Poland; agnieszka.smieszek@upwr.edu.pl (A.S.); klaudia.marcinkowska@upwr.edu.pl (K.M.); mateusz.sikora@upwr.edu.pl (M.S.); 2Institute of Physics, Polish Academy of Sciences, Aleja Lotnikow 32/46, PL-02668 Warsaw, Poland; jablo@ifpan.edu.pl (K.L.-J.); bwitkow@ifpan.edu.pl (B.S.W.); kuzmiuk@ifpan.edu.pl (P.K.); godlew@ifpan.edu.pl (M.G.); 3International Institute of Translational Medicine, Jesionowa 11 Street, 55-124 Malin, Poland; 4Collegium Medicum, Institute of Medical Science, Cardinal Stefan Wyszynski University (UKSW), Wóycickiego 1/3, 01-938 Warsaw, Poland

**Keywords:** atomic layer deposition, titanium dioxide, ultrathin layers, oxide layers, TiO_2_ coating, improved viability, osteogenic properties

## Abstract

Atomic layer deposition (ALD) technology has started to attract attention as an efficient method for obtaining bioactive, ultrathin oxide coatings. In this study, using ALD, we have created titanium dioxide (TiO_2_) layers. The coatings were characterised in terms of physicochemical and biological properties. The chemical composition of coatings, as well as thickness, roughness, wettability, was determined using XPS, XRD, XRR. Cytocompatibillity of ALD TiO_2_ coatings was accessed applying model of mouse pre-osteoblast cell line MC3T3-E1. The accumulation of transcripts essential for bone metabolism (both mRNA and miRNA) was determined using RT-qPCR. Obtained ALD TiO_2_ coatings were characterised as amorphous and homogeneous. Cytocompatibility of the layers was expressed by proper morphology and growth pattern of the osteoblasts, as well as their increased viability, proliferative and metabolic activity. Simultaneously, we observed decreased activity of osteoclasts. Obtained coatings promoted expression of Opn, Coll-1, miR-17 and miR-21 in MC3T3-E1 cells. The results are promising in terms of the potential application of TiO_2_ coatings obtained by ALD in the field of orthopaedics, especially in terms of metabolic- and age-related bone diseases, including osteoporosis.

## 1. Introduction

The coating of surgical implants is designed to improve their biocompatibility and bioactivity. The promotion of bone healing and the restoration of tissue homeostasis are essential factors to be considered when designing new coatings for bone regeneration. Much attention is paid to novel modified coatings with improved biological activity that affects the metabolism of progenitor cells by enhancing their viability and proliferation, as well as supporting cellular adhesion and increasing cellular differentiation. This aspect is crucial, especially in relation to bone metabolic disorders, such as osteoporosis. A thorough analysis of cells’ response to a biomaterial surface can provide some insight into the cellular mechanisms controlling bone metabolism and homeostasis. Insufficient integration of bone tissue with the implant surface can cause the implant to be rejected, leading to severe complications, most often requiring revision surgeries [1]. Thus, tailoring biocompatibility of biomaterials is usually associated with modifying its surface in order to improve the cells’ adhesion, proliferation, and tissue-specific differentiation. Mechanical modifications are aimed at introducing changes in the material’s topography to achieve optimal cell adhesion to the surface, while chemical methods, for example those associated with anodising the titanium surface, yield nanotubes that improve the biological and anti-microbial properties of the biomaterial [2]. Furthermore, sol-gel methods were shown to improve the corrosion resistance of metal substrates and to enhance the osteogenic differentiation of progenitor cells [3,4]. In addition, magnetron sputtering is a useful tool for biofunctionalization of implant surfaces. However, it recently has been shown that the TiO_2_ obtained by ALD technology provides better anti-corrosion properties independent of surface topography in comparison to sputtered TiO_2_ [5]. It is also possible to obtain the high quality layer of TiO_2_ with a Pulsed Laser Deposition (PLD) technique. PLD allows for depositing the films with variable porosity and density [6].

Recently, there has been a growing interest in the application of atomic layer deposition (ALD) technology for tissue-engineering applications and improving implants’ surfaces. ALD technology allows us to fabricate ultrathin, highly uniform, and reproducible coverings with a broad range of potential biological applications thanks to their biomimetic features [7]. Additionally, conformal growth provides the possibility for functionalising the multidimensional surfaces [8]. Furthermore, ALD technology can be used for temperature sensible materials that support tissue regeneration, e.g., polymer-based scaffolds, or composites. It was reported that homogenous ALD layers can be created even at room temperature, which significantly extends their application in terms of scaffold modification and the functionalisation of biomolecules, as well as other temperature-sensitive nanoparticles [9]. For instance, ALD technology was previously used for the deposition of titanium dioxide (TiO_2_) coverings onto tobacco mosaic virus (TMV) and ferritin. The application of ALD yielded pores and channels with diameters less than 4 nm on the TMV, while nanotubes were fabricated with ferritin molecules [10].

The applicability of ALD technology is emerging in the field of bioengineering and regenerative medicine, especially in the light of the significant advantages of this method over other techniques used for depositing oxide coatings onto sensitive substrates. Previously, ALD technology was used to obtain uniform TiO_2_ coatings with controllable thickness, not exceeding 2 nm. The coatings were deposited on porous materials and three-dimensional objects, which indicated their high potential for application in the functionalisation of implants designed for dentistry and orthopaedics [11]. However, Liu et al. for the first time used ALD technology to obtain TiO_2_ nano-coatings with anti-bacterial properties and high bioactivity. Their study showed a wide range of anti-microbial efficacy of ALD TiO_2_coatings that inhibit the growth of gram-positive bacteria (*S. aureus*), Gram-negative bacteria (E. coli), and antibiotic-resistant bacteria (MRSA). Moreover, it was shown that ALD TiO_2_ coatings have a potential selective function, promoting osteoblasts while suppressing fibroblast adhesion and proliferation. This feature is of the utmost importance for orthopaedic implants that are designed to minimise fibrous tissue formation and simultaneously maximise the formation of functional bone tissue [12].

In addition, a study by Yang et al. indicated further potential biomedical application of TiO_2_ coatings, showing that these nano-layers can be deposited on Mg-Zn alloy stents to enhance human coronary artery endothelial cell adhesion and growth. The study revealed that an optimised processing temperature control of the ALD TiO_2_ coatings is essential in order to achieve the proper biological function of the biomaterial. Yang et al. have indicated that coatings deposited at 150 °C have a greater potential to promote the proliferation of endothelial cells than coatings deposited at 200 °C [13]. Furthermore, Basiaga et al. indicated that the mechanical properties of TiO_2_ coatings obtained by ALD technology strictly depend on the number of cycles during the process of deposition [14]. This information is also of practical importance in terms of the application of ALD coatings for designing implants, both for bone regeneration as well as for contact with blood, such as coronary stents. Recently, Motola et al. showed the possibility to enhance the functionality of the Ti surface. They investigated flat and nanotubular interfaces modified with ALD processes and considered the influence of osteoblast, fibroblast, and neuroblast cells’ growth and proliferation [15]. Thin TiO_2_ coatings obtained by photocatalytic patterning can also be improved by flower-like hierarchical Au structures that promote the adhesion and proper growth of the osteoblast cells [16]. Moreover, the TiO_2_ coatings covered with Au nanoparticles may significantly improve their photocatalytic activity [17] and increase their potential application for example as a sensitive detectors of 17β-estradiol [6].

The response of cells to the contact surface is induced with both chemical and physical properties. Wettability, roughness, and isotropic qualities of the materials are an important factor which can determine cell response to the solid state surface and should be taken into account when designing multifunctional coatings [18]. Mendonca et al. analyze the influence of nanoscale roughness and chemical composition on the osteogenesis gene expression [19]. Given the emerging importance of ALD technology in preparing TiO_2_ coatings for contact with bone tissue, we aimed to obtain TiO_2_ coatings that regulate the activity of both bone-forming and bone-resorbing cells. In the current research, we were able to produce homogenous coatings at low temperatures with a thickness of 90 nm. We have investigated their cytocompatibility using a model of mice pre-osteoblasts (MC3T3-E1 cell line) as well as a co-culture with pre-osteoclasts (i.e., 4B12 cell line). We have established the influence of TiO_2_ coatings obtained by ALD and MC3T3-E1′s viability, proliferative potential, metabolic activity, morphology, and growth pattern. In the co-culture model of osteoblasts and osteoclasts, we have also evaluated the influence of TiO_2_ coatings on the expression of markers associated with bone metabolism. The biomarkers were evaluated at the messenger ribonucleic acid (mRNA) and micro ribonucleic acid miRNA levels. This study shows, for the first time, the modulatory effect of TiO_2_ coatings obtained by ALD on the osteoblast–osteoclast coupling.

## 2. Materials and Methods

### 2.1. Substrate

The thin TiO_2_ films were deposited on glass coverslips 13 mm in diameter for biological testing and physicochemical characterisation. The adhesion and quality of the layer deposited by the ALD may be limited by possible surface contamination. This is why all the substrates were thoroughly washed in an ultrasonic cleaner and then dried before the ALD process. The first bath took place in isopropanol, the next two in deionised water. All wash cycles were carried out for 5 min in a temperature bath of 37 °C. Subsequently, nitrogen gas with a purity of 5.0 was used to dry the substrates after the cleaning process.

### 2.2. ALD Growth Method

The deposition of TiO_2_ film was carried out in a Savannah-100 Cambridge NanoTech (now Veeco) reactor. Two precursors were alternately introduced into the reaction chamber. First, an organic precursor was selected as a metal precursor: Tetrakis(dimethyloamino)titanium (CAS no.: 3275-24-9, Strem Chemicals, Inc., Newburyport, MA, USA). The second one, an oxygen precursor, was deionised water. The purging phase after each dose of precursors was carried out with nitrogen gas with a purity of 6.0. The precursor feeding was as follows: 0.2 s dose of metal precursors, 3 s waiting phase,15 s pulse of purging gas, 0.04 s pulse of oxygen precursor, 3 s waiting phase, and another purging gas dose of 15 s. This protocol was repeated 1220 times. The process was carried out under stable temperature (100 °C) and pressure (66 Pa). Additionally, the titanium precursor was preheated to 70 °C. The heaters of the reactor chamber and precursor were turned on an hour before the start of the process to ensure that the correct temperature was reached and stabilised and held during the layer growth. After ALD processes, the samples were vacuum packed and transferred to further biological and physical research. There were no additional cleaning procedures before the physical measurements.

### 2.3. Analysis of the Physicochemical Properties of the Coatings

The X-ray diffraction (XRD) assays were used to determine the crystallinity of the TiO_2_ films. The panalytical X’Pert Pro MRD diffractometer was used in the XRR analysis. The generated X-ray radiation was at a wavelength of 1.54056 Å. The Pixcel detector and Parallel Plate Collimator with 0.4-rad Soller slits and a 0.18-deg divergence slit were applied. Based on Parratt’s theory, Panalytical software [20] was used to determine the thickness, electron density, and roughness of the resulting coating. The rate of decay of the X-ray signal can determine the roughness of the surface: the amplitude of the oscillations observed is related to the thickness of the layer on the surface of the substrate and the width of the oscillations changes with the electrons’ density of the material.

The Scanning Electron Microscopy (SEM) investigations were performed using a Hitachi SU-70 system. Images were taken at 15 kV of accelerating voltage using detector of secondary electrons.

To determine the chemical compounds of the layers obtained, X-ray photoelectron spectroscopy (XPS) measurements were taken. The XPS measurements were carried out according to a previously described protocol [21] using a Scienta R4000 hemispherical analyser (pass energy: 200 eV) and Al K_α_ (1486.7 eV) with non-monochromatic excitation. The analysis of full width at half maximum (FWHM) of the 4f7/2 Au line was measured under the same experimental conditions as TiO_2_ layer was 1.1 eV. The energy scale was calibrated by setting the C 1s line at the position of 285.6 eV [21].

The wettability of the surface is a crucial parameter in determining the biocompatibility of the material. We assessed the hydrophilic/hydrophobic properties of biologically tested substrates by measuring the contact angle of a drop of water deposited on the surface. A goniometer OCA 25 from DataPhysik was utilised to obtain the contact angle for fluid on the TiO_2_ surface deposited onto a coverslip. The test fluid was water. The measurement was performed under normal conditions (temperature: 25 °C; air humidity: 50%). As a reference, the wettability measurement on an uncoated coverslip was taken. All tests were repeated three times at different sites on the samples, a TiO_2_-coated coverslip, and an uncoated coverslip. The result for links and rights contact angle was taken into account.

### 2.4. Evaluation of the Cytocompatibility of TiO_2_ Coatings Obtained by ALD

Culture of pre-osteoblastic mice cell line: MC3T3-E1 cell line cells were maintained in Minimum Essential Media Alpha (MEM-α, Gibco™ Thermo Fisher Scientific, Warsaw, Poland). The culture conditions have been thoroughly described by other authors [21,22]. The medium was supplemented with 10% Fetal Bovine Serum, (FBS, Sigma Aldrich, Munich, Germany) and changed every 2–3 days. The cells were cultured at constant, aseptic conditions in a CO_2_ incubator at 37 °C and 95% humidity. The cultures were passaged using a trypsin solution (StableCell Trypsin, Sigma Aldrich, Munich, Germany) after reaching 90% confluence. The protocol was previously described in detail [22]. Before the trypsinisation step, the cells were washed in Hanks’ Balanced Salt Solution (HBSS) without calcium or magnesium. Th cells were passaged using trypsin solutions and common protocols [22]. For the digestion, 3 mL of trypsin solution was added to a T75 culture flask (Nunc, USA), and the cells were incubated for 5 min at 37 °C in a CO_2_ incubator. The cells used for these experiments were at passage number 24 (p24). For the experiment, MC3T3-E1 cells were inoculated in 24-well plates at a density equal to 2 × 10^5^ cells per well. The cells were maintained in 0.5 mL of CGM (MEM-α with 10% FBS). The medium was refreshed 3 times per week.

The influence of TiO_2_ coatings obtained by ALD on the apoptosis profile: The apoptosis profile was analysed using a MUSE Annexin V & Dead Cell Kit (Merck, Sigma-Aldrich, Poznan, Poland) according to the manufacturer’s protocols and previously published information [22]. To put it briefly, cells were detached with trypsin, centrifuged (5 min, 300× *g*), and diluted with 100 μL of PBS containing 1% FBS. The cell suspension and 100 μL of Annexin V/dead reagent were mixed in a 1.5-mL centrifuge tube and incubated for 20 min at room temperature in the dark. Then, the cells were analysed using a Muse Cell Analyser (Merck, Sigma-Aldrich, Poznan, Poland). The apoptotic cell distribution was evaluated by identifying four populations: non-apoptotic (viable) cells: Annexin V (−), 7-AAD (−); early apoptotic cells: Annexin V (+), 7-AAD (−); late apoptotic/dead cells: Annexin V (+), 7-AAD (+); and dead cells: Annexin V (−) and 7-AAD (+).

The influence of TiO_2_ coatings obtained by ALD on cell proliferation: The proliferative activity was evaluated using a MUSE Cell Cycle Kit (Merck, Sigma-Aldrich, Poznan, Poland) according to the manufacturer’s instructions. A detailed protocol was published by other authors [22]. For the analysis, the cells were collected after detachment from the culture flask with trypsin solution, centrifuged (5 min, 300× *g*), washed with 1X PBS, fixed with 1 mL of 70% cold ethanol, and incubated overnight. Then, the cells were centrifuged and washed, as described previously [22]. The cell pellet was suspended in 200 μL of Muse Cell Cycle Reagent (Merck, Warsaw, Poland) and incubated at room temperature for 30 min in the dark. The distribution of cells in G0/G1, S, and G2/M phases was estimated using a Muse Cell Analyser (Merck, Sigma-Aldrich, Poznan, Poland).

The influence of TiO_2_ coatings obtained by ALD on the cells’ metabolic activity: The metabolic activity of the cells was estimated using a TOX-8 resazurin-based in vitro Toxicology Assay Kit (Sigma Aldrich, Munich, Germany) according to the manufacturer’s instructions and previously published protocols [23]. The complete growth medium was replaced with fresh CGM supplemented with 10% resazurin solution (Sigma Aldrich, Munich, Germany). The cells were incubated for 2 h at 37 °C in a CO_2_ incubator. Then, the supernatants were transferred to a 96-well plate (100 μL per well). The absorbance was measured as indicated before [4,21,22]. The effect of TiO_2_ coatings on the cells’ metabolic activity was assessed after 24, 48, 72, 96, and 168 h of culture (not all data are shown).

The influence of TiO_2_ coatings obtained by ALD on mitochondrial potential: changes in the mitochondrial potential of MC3T3-E1 cells cultured in TiO_2_ were monitored using a MUSE MitoPotential Kit (Merck, Warsaw, Poland) according to the supplier’s protocols. The cells were harvested by trypsinisation, centrifuged (5 min, 300× *g*), and suspended in 100 μL of Assay Buffer (Merck, Warsaw, Poland). Next, 95 μL of MitoPotential working solution was added to the cell suspension. Assay tubes were vortexed for 3 s and incubated for 20 min in a CO_2_ incubator at 37 °C. Then, 5 μL of Muse MitoPotential 7-AAD reagent (Merck, Warsaw, Poland) was added to the cell suspension and the samples were incubated for 5 min at room temperature. The mitochondrial potential was assessed using a Muse Cell Analyser (Merck, Warsaw, Poland) by the identification of four populations: live cells with depolarised mitochondrial membrane: MitoPotential (-), 7-AAD (-); live cells with intact mitochondrial membrane: MitoPotential (+), 7-AAD (-); dead cells with depolarised mitochondrial membrane: MitoPotential (+), 7-AAD (+); and dead cells with disturbed mitochondrial membrane potential: MitoPotential (-) and 7-AAD (+).

Determination of the TiO_2_ coatings’ obtained by ALD influence on cell morphology, ultrastructure, and adhesion rate: The morphology and ultrastructure of the MC3T3-E1 cells were analysed after 72 h of culturing. The detailed protocol of culture staining and preparation for confocal imaging was described by other authors [24]. The specimens were analysed using a confocal microscope (Leica TCS SPE, Leica Microsystems, KAWA.SKA Sp. z o.o., Zalesie Gorne, Poland) and the microphotographs obtained were then analysed using Fiji (ImageJ 1.52n, Wayne Rasband, National Institute of Health, Bethesda, Maryland, USA), as described previously [22]. The figures presented herein were obtained using the maximum intensity projection (Z-projection). In addition, the ultrastructure of cultures was examined with a scanning electron microscope (SEM, Zeiss Evo LS 15, Oberkochen, Germany). Before SEM analysis, the cells were fixed (in 4% paraformaldehyde [PFA] as described above) and dehydrated in an ethanol series (concentrations from 50% to 100%, each incubation for 5 min). The specimens were sputtered with gold and observed using an SE1 detector at 10 kV of filament tension [4]. The adhesion rate of the MC3T3-E1 cells was determined using the protocol published by Huang et al. [25] and used previously [26,27].

Co–culture with pre-osteoclastic cell line 4B12: The osteoclast precursor cell line 4B12 was kindly provided by Shigeru Amano from the Department of Oral Biology and Tissue Engineering, Meikai University School of Dentistry. The detailed description of the culture method was published before [22]. The cells used in the experiment were at passage number 28 (p28). The 4B12 cells were cultured with pre-seeded MC3T3 cells. For co-culturing, the 4B12 cells were inoculated at a density equal to 3.5 × 10^4^ in a chamber of an 8-µm Transwell system membrane (Corning, Biokom, Warsaw, Poland). The cells were maintained in 0.3 mL of α-MEM with 10% FBS and 30% CSCM; half of the medium was changed 3 times per week. Th edetailed protocol of co-culturing was described elsewhere [22]. The invasion of 4B12 was determined based on SEM images.

The influence of TiO_2_ coatings obtained by ALD on osteogenesis marker gene expression and miRNA levels: To determine the mRNA and miRNA levels, the experimental cultures were homogenised using 1 mL of Extrazol^®^ (Blirt DNA, Gdansk, Poland). The protocol for total RNA isolation was performed according to the manufacturer’s instructions and the modified phenol-chloroform method described by Chomczyński and Sacchi [28]. The resulting RNA was diluted in DEPC-treated water. The quantity and purity of RNA specimens were determined spectrophotometrically at 260- and 280-nm wavelengths (Epoch, Biotek, Bad Friedrichshall, Germany). Before reverse transcription, the total RNA obtained (500 ng) was purified using DNAse I (PrecisionDNAse, PrimerDesign, BLIRT S.A. Gdansk, Poland). The reverse transcription was performed using a Tetro cDNA Synthesis Kit (Bioline Reagents Limited, London, UK). Both processes were performed according to well-established protocols [14,15]. The DNA digestion and cDNA synthesis were carried out in a T100 Thermal Cycler (Bio-Rad, Hercules, CA, USA). The resulting matrices were used for RT-qPCR analysis using a SensiFAST SYBR^®^&Fluorescein Kit (Bioline Reagents Ltd., London, UK). The final reaction volume was 10 ul, where 1 ul of cDNA was used and the concentration of primers was 0.5 µM. Quantitative PCR was performed in a CFX Connect Real-Time PCR Detection System (Bio-Rad, Hercules, CA, USA). The details of the protocols and the reaction conditions were described previously [22]. Additionally, to evaluate the miRNA levels, cDNA was also synthesised using 375 ng of total RNA with a Mir-X™ miRNA First-Strand Synthesis Kit (Takara Bio Europe, Saint-Germainen, Laye, France) as described by other authors [21]. The primer sequences are summarised in Appendix A. All qPCR reactions were carried out in at least three repetitions. The relative values of gene expression were determined with the RQ_MAX_ algorithm as described previously [29]; for normalisation, Gapdh (glyceraldehyde 3-phosphatedehydrogenase) was used as a reference gene and U6snRNA (Takara Bio Europe, Saint-Germainen, Laye, France) was used to determine the levels of miRNA.

### 2.5. Statistical Analysis

The results obtained during in vitro studies are presented as the mean of a minimum of three trials. The means are presented with standard deviations (± SD). Statistical comparisons between the data were assessed by one-way analysis of variance (ANOVA) and unpaired Student’s *t*-test. The results were analysed using GraphPadPrism 5 software (La Jolla, CA, USA). Differences with a probability of *p* < 0.05 were considered to be statistically significant.

## 3. Results

### 3.1. Physicochemical Properties of TiO_2_ Coatings Obtained by ALD

The thin layer of titanium dioxide was successfully deposited onto glass substrates for biological investigations and physicochemical tests.

The XPS measurement of O 1s, C 1s, and Ti 2p lines correspond to a content of 22.1 at.% of titanium (Ti), 52.3 at.% of oxygen (O), and 25.6 at.% of carbon (C) on the sample surface. No other elements were found (Figure 1a). The relatively high adventitious carbon originated mostly from contamination of the surface due to air exposition of the sample before XPS measurements and was used for energy scale calibration (C 1s binding energy [BE] 285.6 eV). The stoichiometry ratio in the oxide compound is determined by the content of the metal ions and the oxygen ions. The ratio of O:Ti content was 2.4, confirming the growth of an amorphous layer with stoichiometry close to TiO_2_. The excess oxygen may have resulted from the adsorption of water and carbon oxide on the surface. The result was confirmed by analysis of the oxygen 1s line Figure 1b). Two components fit this line well: the main component at BE = 530.8 eV covers 86% of the line and the second component BE = 532 eV covers only 14%. Taking into account only the main component at 530.8 eV of energy—which is close to the oxygen-binding energy in TiO_2_ [30,31]—the corrected ratio of oxygen to titanium is 2.04. It confirms the formation of an amorphous titanium dioxide with stoichiometry close to the ideal. The energy positions (459 eV and 464 eV) and the separation in the Ti 2p spin-orbit doublet (5.6 eV) agree within the limit of energy calibration error (± 0.2 eV) with the values reported in the reference table and other publications (Figure 1c) [32]. Moreover, in the case when the many O defects are present and stoichiometry of ALD layer is strongly changed, the presence of Ti^+3^ is observed in Ti 2p line as e.g., in [5].

The XRR measurement was performed on the coverslip deposited with the TiO_2_ coating as well as on the coverslip. We observed a change in the surface roughness of the samples. The simulation of experimental data indicates that the roughness of the surface of the coverslip is 0.7 nm (Figure 2b); for the TiO_2_ coating, the roughness parameter was simulated to the value of 2.4 nm (Figure 2c). The simulated density is equal to 3.88 g/cm^3^, which is lower than the table data from bulk 4.23 g/cm^3^ titanium dioxide material. The thickness of the deposited coating was estimated to 90 nm.

To determine the quality of the coating, we performed SEM measurements. The images presented in Figure 3 show a very high uniformity (Figure 3a) of the coating. No larger crystallites are visible on the surface, which is characteristic of amorphous layers (Figure 3a,b).

Additionally, to confirm the amorphous phase of the coating, we performed X-ray diffraction measurements in the Theta/2Theta configuration. We did not detect any additional signal in comparison to the broad peak of an uncoated cover glass (Appendix A)**.** A broad peak from the substrate confirmed the amorphous nature of glass, and the absence of a signal from the layer also points to an amorphous structure. The obtained result is in good agreement with our previous report [33] and other research [9].

The evaluation of the wettability data (Figure 4) shows that, in relation to the surface of the coverslip (Figure 4a), the TiO_2_ sample (Figure 4b) has a much higher value of the contact angle. The value of the contact angle is for coverslip 62.9° and 93.3° for TiO_2_ sample. This finding indicates a more substantial hydrophobic property of TiO_2_ compared to a pure coverslip.

### 3.2. Cytocompatibility of the TiO_2_ Coatings Obtained by ALD

The analysis of cell viability indicated that the TiO_2_ coating exhibited non-toxic properties toward the MC3T3-E1 cell line (Figure 5). The biomaterial significantly improved the viability of preosteoblasts (Figure 4a,b). Moreover, we observed decreased apoptosis in MC3T3-E1 cultures propagated on TiO_2_ obtained by ALD, though the differences were not statistically significant (Figure 5c).

The analysis of the distribution of cells in the cell cycle revealed that preosteoblast MC3T3-E1 cells cultured on TiO_2_ surfaces obtained by ALD (Figure 6) showed increased proliferative activity, which was reflected in the accumulation of cells in the S phase (Figure 6a,c). Moreover, in cultures propagated on TiO_2_ coatings obtained by ALD, we observed more cells in the G2/M phase (Figure 6a,d). The shift of cells toward the G2/M phase was accompanied by fewer cells in the G0/G1 phase (Figure 6a,b).

The Alamar Blue test indicated increased metabolic activity of MC3T3-E1 cells cultured in the presence of TiO_2_ coatings (Figure 7). The MC3T3-E1 cells demonstrated a significantly faster metabolism in response to TiO_2_ coating obtained by ALD at the initial stage of culturing, i.e., after 24 h of propagation (Figure 7a), but also after 72 h and 168 h of culturing (Figure 7c,e).

The analysis of cells morphology showed no significant influence of TiO_2_ coating on the cells’ morphology or mitochondrial network development (Figure 8a). Our observations revealed that both the control cultures and the experimental cultures had a properly expanded network of the actin cytoskeleton, maintained cell–cell contact, and appropriately adhered to the substrate. Moreover, both the MC3T3-E1 cells cultured on a plain coverslip and those coated by ALD with TiO_2_ had well-developed mitochondrial networks (Figure 8a). The MC3T3-E1 cells maintained proper morphology and ultrastructure in both the control and experimental cultures. In addition, no occurrence of apoptotic bodies was observed in the MC3T3 cultures (Figure 8b). Nevertheless, we found that MC3T3-E1 cells propagated onto TiO_2_ surfaces were characterised by lower mitochondrial potential in comparison to the control cultures; this difference was statistically significant (Figure 8c,d). At the same time, we did not observe significant differences in the percentage of total depolarised cells between the control and experimental cultures propagated on TiO_2_ layers (Figure 8c,e).

The analysis of preosteoclast invasion properties indicated that the number of preosteoclasts is significantly lower in the cultures of the preosteoblast MC3T3 propagated on a TiO_2_ coating, (Figure 9a,b). In addition, the measurement of MC3T3-E1 adhesion indicated that TiO_2_ coatings promote osteoblast attachment to surfaces. The results are in agreement with the increased metabolic activity of MC3T3-E1 preosteoblasts propagated on a TiO_2_ layer, measured when they first interact with the biomaterial, which strictly depends on the adhesive properties of the surface.

The pro-osteogenic properties of the TiO_2_ coatings were confirmed by the expression of genes involved in the process of osteogenesis and proper bone mineralisation. The analysis was performed for the MC3T3-E1 cell line, as well as for co-cultures of MC3T3-E1 with pre-osteoclastic cell line 4B12. In the MC3T3-E1 cultures propagated on TiO_2_ surfaces, we observed a higher expression of late osteogenesis markers, such as osteopontin (*Opn*) and osteocalcin (*Ocl*) (Figure 10b,d). Simultaneously, the same cultures were characterised by lower levels of mRNA for other osteogenic markers, i.e., collagen type 1 (*Coll-1*) and runt-related transcription factor 2 (*Runx2*) (Figure 10a,c). Interestingly, the profile of osteogenic markers for MC3T3-E1 cultured with 4B12 was maintained in the cultures propagated on the TiO_2_ coating. As a result of the paracrine effects of preosteoclasts, MC3T3-E1 cultured in control conditions had lower levels of osteogenic genes. Obtained results correspond with the increased invasiveness of 4B12 noted in the control cultures. In turn, the MC3T3-E1 propagated on TiO_2_ and in direct contact with 4B12 were characterised by a higher accumulation of transcripts for *Opn, Ocl*, and *Runx2* (Figure 9b,c and Figure 10a–d).

The levels of microRNAs involved in bone metabolism were also altered in response to TiO_2_ coatings (Figure 11). In the MC3T3-E1 cells cultured onto TiO_2_ obtained by ALD surfaces, we observed a significantly higher expression of miR-17 (Figure 11c) and miR-21(Figure 11d), while miR-124 (Figure 11b) levels were lowered. This profile corresponds with mRNA levels for osteogenic markers, i.e., *Opn* and *Ocl*, showing that the TiO_2_ layer may provide pro-osteogenic conditions, inducing differentiation of MC3T3-E1 cells into osteoblasts. However, the MC3T3-E1 cells propagated on ALD covered with TiO_2_ samples and influenced by the paracrine activity of preosteoclasts showed significantly lower levels of miR-7 and miR-21, which are considered osteogenic miRNAs. Nevertheless, the lower levels of miRNAs promoting osteoclast activity, i.e., miR-7 and miR-124, were significantly lower, a finding which also correlates with the decreased invasion of 4B12 preosteoclasts in this condition.

## 4. Discussion

Currently, much interest can be observed in the development and application of ALD technology as a method for creating bioactive coatings for orthopaedic implants. Mounting evidence indicates that ALD technology provides a new option for functionalising the biomaterials’ surface, improving the metabolism of bone progenitor cells, and promoting osseointegration. In this study, we used ALD technology to obtain thin TiO_2_ films distinguished by selective biological properties: activating pro-osteogenic signals and inhibiting the invasion of osteoclast precursors. TiO_2_ coatings had previously been described and characterised in terms of their antibacterial properties and their cytocompatibility toward progenitor cells, including preosteoblasts [1,2]. It was indicated that thin, ALD-fabricated TiO_2_ meets the criteria of pro-osteogenic coatings that promote bone-forming cell growth and proliferation [12]. The quality of the TiO_2_ layers obtained by ALD meets the requirements of coatings for implant materials. A thin TiO_2_ layer was deposited on scaffolds made of titanium powder, mimicking the biological functions of the substrate. The TiO_2_ was coated on the porous metallic biomaterial uniformly and with high quality [11]. Additionally, TiO_2_ films deposited on 316 LVM steel surfaces have been investigated mechanically. Basiaga et al. demonstrated that modifying the surface of vascular stents is possible; the mechanical properties of such layers—the thicknesses of the layers—depend on the number of ALD cycles used in the process [14].

The ALD technique attracts attention as a promising technology that allows for tailored, unique biocompatible coatings to be fabricated. However, the TiO_2_ coatings obtained by ALD and designed for potential biomedical application are usually created in temperatures significantly higher than 100 °C. For example, Liu et al. deposited TiO_2_ at 200 °C [34], while Liu et al. created coatings with ALD in the 120–190 °C temperature range [35]. In this study, we performed the TiO_2_ growth process at a temperature of 100 °C. The selection of precursors, the metal precursor in particular, and the temperature during the ALD growth process are the key parameters which determine the phase composition of the thin TiO_2_ coating. The tetrakisdimethyloamino used by us in this study has a low growth rate of TiO_2_, indicating inefficient surface reactions. For comparison, the growth rate when using tetrakisdimethyloamino of metals such as hafnium or zirconium is twice as high [33]. However, despite the low growth rate, the high quality of the coating indicates the stoichiometry of this dioxide was preserved. TiO_2_ occurs in various crystallographic phases—amorphous, rutile, anatase, and brookite—or it can coexist in several phases [35]. The crystal phase of TiO_2_ obtained by ALD depends strongly on the deposition temperature [9], while the amorphous layer is formed at low deposition temperatures, anatase at medium temperatures, and rutile at the highest temperatures. It has also been proven that crystallography influences the biological properties of the coating. For example, Rossi et al. have shown that TiO_2_ coating which contains an anatase and rutile phase absorbing the proteins from physiological fluids better [36]. The initial protein adhesion to the surface determines the developmental phases of the cells (differentiation and proliferation) and whether osseointegration is successful. Despite a lack of noticeable crystallographic order of the Ti and O atoms, we found excellent osteogenic properties of the TiO_2_ coating.

An amorphous structure leads to better adhesion to the substrate as compared to the corresponding crystalline layer [33]. Moreover, a low temperature of deposition is a significant advantage, allowing such a coating to be applied on polymer surfaces that can change their structure at high temperatures. Amorphous oxides obtained by low temperature are often far from stoichiometric. Park et al. showed in their work the formation of crystallites with higher oxygen content while depositing TiO_2_ with plasma-enhanced ALD. While the TiO_1.6_ layer was amorphous, the increased oxygen content (TiO_1.7_) resulted in the formation of crystallites [37]. In our study, we obtained amorphous TiO_2_ coating close to ideal stoichiometry. The reason for this phenomena can be the fact that the generation of oxygen vacancies was thermodynamically blocked at low temperatures during the growth process.

The TiO_2_ coating increased the hydrophobicity of the surface. This result is contrary to data reported by Liu et al. They found a lower water contact angle with the deposition of an ALD layer. However, in such a case, the wettability was determined by the substantial increase of surface roughness rather than the surface chemistry. Cell adhesion is generally strongly correlated with the hydrophilic properties of the materials. We found better cell adhesion despite the higher water contact angle. It appears that their finding depended on the surface chemistry, and that the nearly ideal stoichiometry may influence the biological properties of TiO_2_.

In this study, we found that 90-nm TiO_2_ coatings obtained by ALD may promote proper bone formation and may enhance the viability, proliferation, and metabolic activity of preosteoblasts. We were able to determine the influence of TiO_2_ coatings obtained by ALD on both osteoblast and osteoclast activity. We have indicated that TiO_2_ layers improve the metabolic activity and viability at the early stages of cell–biomaterial contact and that they lesson the invasion of osteoclast progenitors. These features are extremely desirable and are required for bone implant coverings, as they can ensure the proper integration of biomaterials with bone tissue and can guarantee active bone remodelling. We have also found that TiO_2_ coatings promote the adhesion of preosteoblasts and have the features of a biomimetic structure, allowing for the control of cell–surface interaction.

The improved adhesion of osteoblast to TiO_2_ coverings was noted previously. For example, Shokuhfar et al. reported that Ti surfaces treated with amorphous and crystalline TiO_2_ nanotube are effective in increasing the number of attached MC3T3-E1 preosteoblasts. In addition, using SEM and FIB analysis, Shokuhfaret al. provided direct evidence on the interlocked mechanism between the cell and TiO_2_. It was shown that osteoblasts growing on nanostructured TiO_2_ coatings create filopodia extensions, increasing the contact area and resulting in better anchorage to the surfaces [38]. The increased adhesion of osteoblasts to TiO_2_ coatings was also described by Rivera-Chacon et al., who explained this phenomenon by the selective absorbance of vitronectin and fibronectin by substrates with nanostructures [39]. This finding partially explains our results, which indicate the improved attachment of osteoblasts into the TiO_2_ layer and the inhibited invasion of osteoclasts. Vitronectin was shown to promote osteoblast differentiation and activity, whilst concomitantly restraining osteoclast differentiation and resorptive function [40].

The increased adhesion, as well as the improved proliferation and viability of progenitor cells, ensures the guided regeneration of bone and the formation of functional tissue. In this study, we showed that TiO_2_ coatings exert an anti-apoptotic effect towards preosteoblasts, significantly increasing their viability and promoting cellular metabolism. Such features were described previously in relation to the cytocompatibility of TiO_2_ coatings. TiO_2_ coatings obtained through ALD had been reported as bioactive layers that modulate the metabolism of progenitor cells, affecting their osteogenic potential. In this study, we confirmed that TiO_2_ coatings obtained by ALD activate transcripts associated with preosteoblast differentiation into bone-forming cells. We found that TiO_2_ layers increased the mRNA levels of osteopontin and osteocalcin in the pre-osteoblastic MC3T3-E1 cell line, which is in line with the results presented by Vercellino et al., who showed that titanium dioxide nanostructured coatings promote the differentiation of bone marrow stromal cells, elevating the expression of osteopontin and osteocalcin [41].

In addition, we found that the gene expression pattern correlates with higher levels of regulatory microRNAs, such as miR-17 and miR-21. It was previously reported that TiO_2_-nanotube arrays regulate the miRNA levels in human adipose-tissue-derived stem cells (hASCs) propagated under osteogenic conditions. The increasing interest in microRNA involvement in the regulation of pro-osteogenic signals also reinforces the studies on the effect of biomaterial and nanotopography-guided differentiation of progenitor cells. Understanding mRNA–miRNA networks as an axis regulating the fate of progenitor cells can be paramount when designing biomaterial-based therapies for metabolic disorders, including osteoporosis. For example, it was previously reported by various groups, including ours, that miR-21 promotes osteogenesis, but also acts as a regulator of osteoclastogenesis and a promoter of osteoclast differentiation [22,42]. Similarly, it was indicated that miR-17-5p improves cell proliferation and osteoblastic differentiation of human multipotent stromal cells. Furthermore, it was shown that decreased expression of miR-17-5p is correlated with worse clinical characteristics and poor survival rate in patients with non-traumatic osteonecrosis [43]. Instead, miR-7 levels have not been thoroughly described in terms of osteoblast biology, and it was found that it can be differentially expressed, depending on bone metabolism [44,45]. It seems that the inhibition of miR-7 targeting the epidermal growth factor receptor (EGFR) may inhibit the development of osteoporosis [45]. This conclusion is in agreement with the profile of miR-124, which is an essential molecule regulating osteoclastogenesis. The overexpression of miR-124 could inhibit osteoclastogenic differentiation of bone-marrow-derived monocyte cells, indicating that the inhibition of miR-124 expression might be a potential therapeutic strategy for the treatment of osteoporosis [46].

Pro-osteogenic properties of TiO_2_ layers obtained by ALD can also be expressed by decreased mitochondrial membrane potential. This is the characteristic feature of differentiated MC3TC3-E1 osteoblasts, as described by Guntur et al. Moreover, a lack of a significant increase in mitochondrial volume fraction during the differentiation of MC3T3-E1 cells to osteoblasts was also observed, which can be explained by the fact that differentiated osteoblasts are not programmed to use oxidative phosphorylation to supply their ATP demand [47]. Our results indicated that TiO_2_ coatings obtained by ALD can play the role of a regulator mitochondrial adaptation and can exert anti-apoptotic effects toward osteoblast precursors. This suggests that their potential application in metabolic- and age-related bone diseases.

## 5. Conclusions

Nanoscale and biomimetic TiO_2_ coatings obtained by ALD have displayed promising pro-osteogenic properties, activating the osteogenic biomarkers associated with proper bone remodelling and regulating mitochondrial activity. We demonstrated that TiO_2_ coverings significantly promote the adhesion of preosteoblast cells and inhibit the invasion of preosteoclasts, lowering the levels of microRNAs (miR-7 and miR-124), which are crucial for osteoclast survival and maturation. The TiO_2_ coatings obtained by ALD can be a suitable layer for enhancing the osteogenic properties and biofunctionality of substrates used in the field of orthopaedics, especially in terms of metabolic- and age-related bone diseases.

## Figures and Tables

**Figure 1 materials-13-04817-f001:**
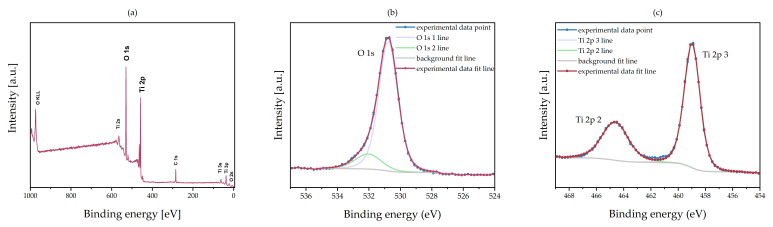
The XPS data plot of TiO_2_ obtained by ALD on a glass substrate: (**a**) wide XPS spectra—only lines of O, Ti, and C were detected, C 1s binding energy (BE) 285.6 eV was used for energy scale calibration; (**b**) O1s experimental and fitting line; (**c**) Ti 2p experimental and fitting line.

**Figure 2 materials-13-04817-f002:**
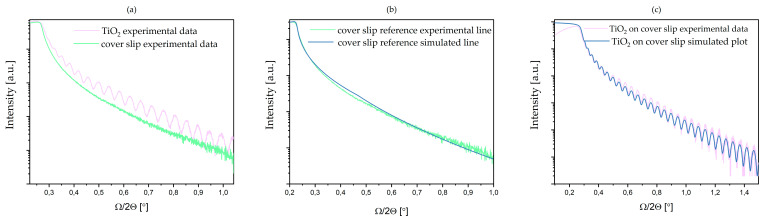
The XRR data plots of TiO_2_ obtained by ALD on a glass substrate (green colour) and uncoated coverslip (magenta colour): (**a**) the experimental line; (**b**) the fitting line (blue colour): for the coverslip sample; (**c**) the fitting line for TiO_2_ on the coverslip sample (fit simulation made using Parratt’s theory).

**Figure 3 materials-13-04817-f003:**
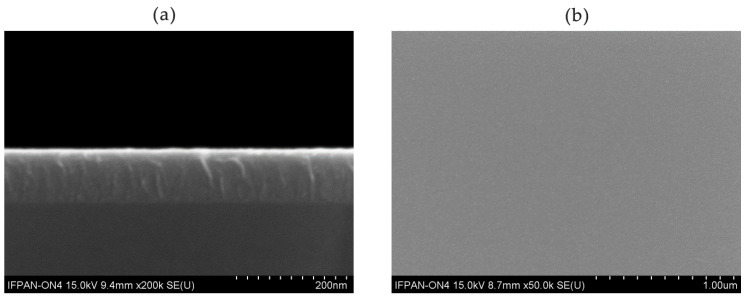
The SEM images of TiO_2_ obtained by ALD on Si substrate as a reference: cross-section view (**a**) and top view (**b**). Images were taken at 15 kV of accelerating voltage using a detector of secondary electrons.

**Figure 4 materials-13-04817-f004:**
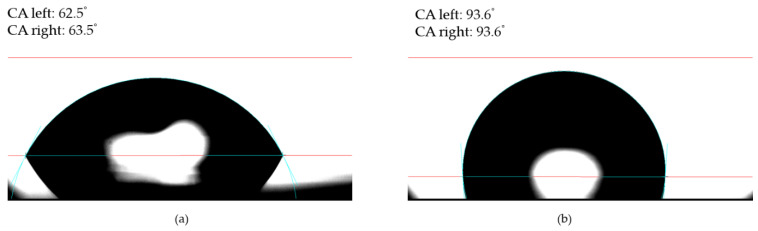
The images of water contact angle (wettability) measurement for (**a**) a coverslip and (**b**) TiO_2_ on a coverslip (CA left indicates measured links’ contact angle, while CA right indicates measured right contact angle).

**Figure 5 materials-13-04817-f005:**
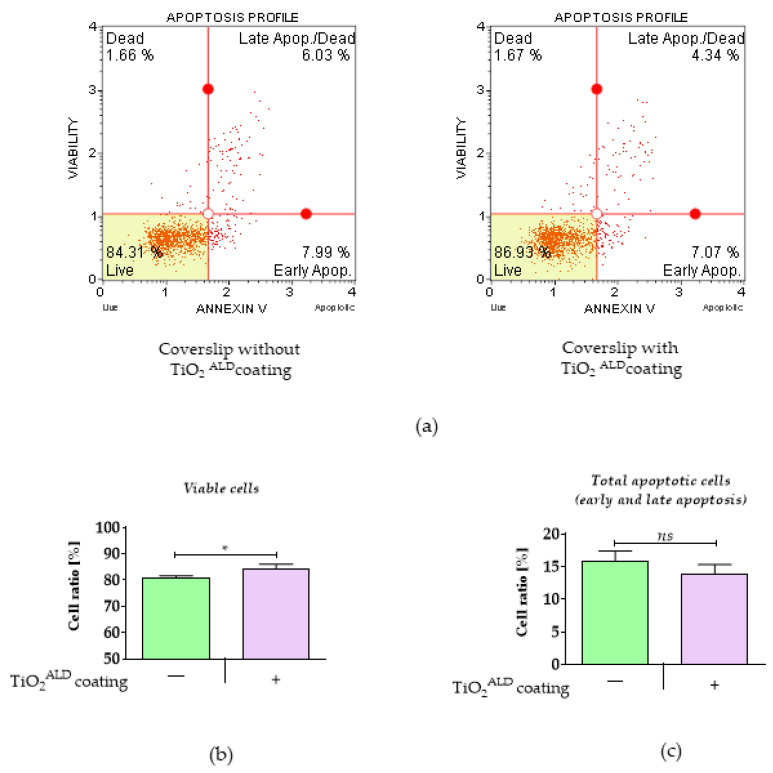
The cell viability and apoptosis profile in the control culture (TiO_2_
^ALD^ coatings-) and the culture propagated on TiO_2_ coatings (TiO_2_
^ALD^ coatings +; experimental culture): (**a**) representative graphs obtained during analysis, showing the distribution of cells on four populations: live (Live—bottom-left corner), early apoptotic (Early Apop.—bottom-right corner), late apoptotic (Late Apop./Dead—upper-right corner), and dead (Dead—upper-left corner); results of statistical analysis showing a comparison of viable cells (**b**) and apoptotic cells (**c**) in control and experimental cultures (significant differences are marked with asterisks (* *p* < 0.05), non-significant results are marked as ns).

**Figure 6 materials-13-04817-f006:**
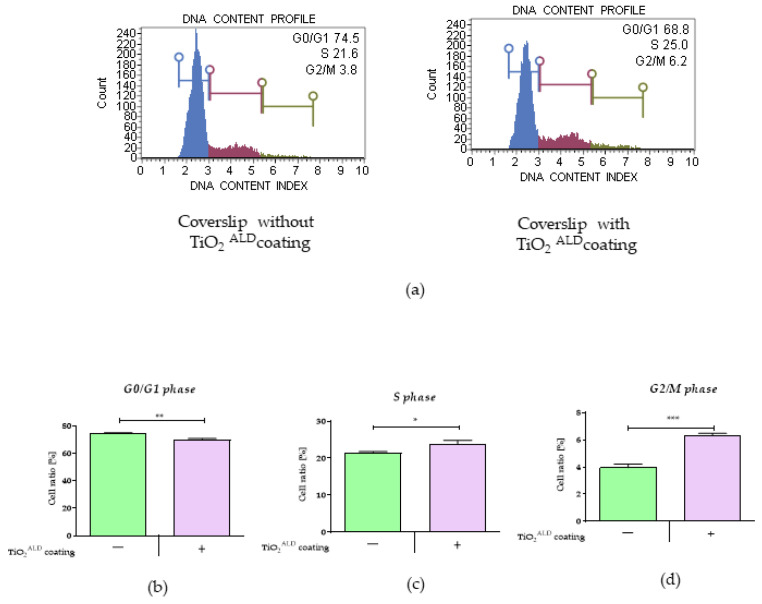
The results of the analysis of the control culture (TiO_2_
^ALD^ coatings-) and a culture propagated on TiO_2_ coatings (TiO_2_
^ALD^ coatings +; experimental culture), showing the distribution of cells in the cell cycle: (**a**) Representative histograms show the distribution of MC3T3 cells in the cell cycle phase under both culture conditions. The cells were separated into three populations: G0/G1 phase (the left side), S phase (the middle), and G2/M phase (the right side). The results of statistical analysis revealed differences in cell distribution during the (**b**) G0/G1 phase, (**c**) S phase, and (**d**) G2/M phase. (significant differences are marked with asterisks (*** *p* < 0.001; ** *p* < 0.01; * *p* < 0.05).

**Figure 7 materials-13-04817-f007:**
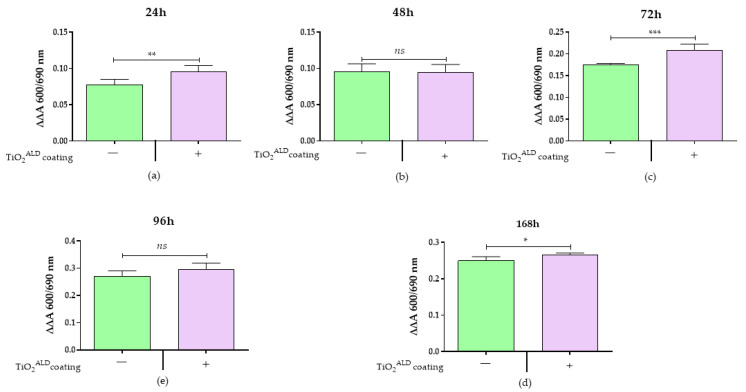
The metabolic activity in the control culture (TiO_2_
^ALD^ coatings-) and a culture propagated on TiO_2_. coatings (TiO_2_
^ALD^ coatings +; experimental culture): comparative analysis of metabolic activity after (**a**) 24 h, (**b**) 48 h, (**c**) 72 h, (**d**) 96 h, and (**e**) 168 h of propagation. (significant differences are marked with asterisks (*** *p* < 0.001; ** *p* < 0.01; * *p* < 0.05), non-significant results are marked as ns).

**Figure 8 materials-13-04817-f008:**
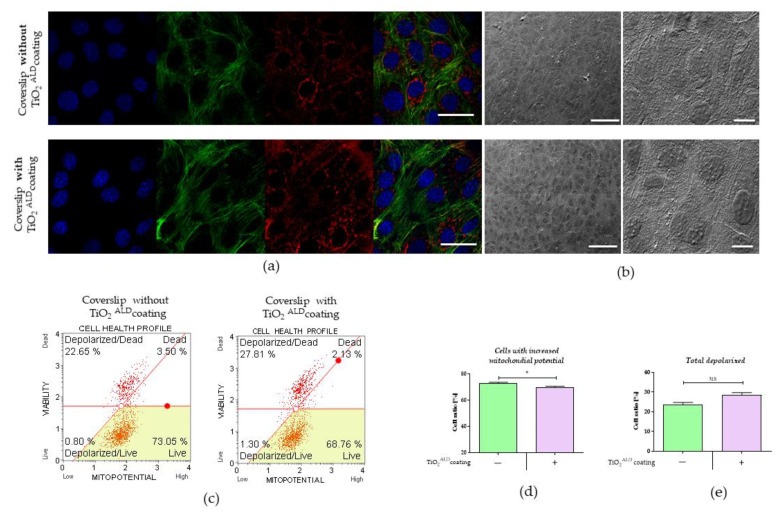
The results of the ultrastructural analysis of MC3T3 cells in the control culture (TiO_2_
^ALD^ coatings -) and a culture propagated on TiO_2_ coatings obtained by ALD (TiO_2_
^ALD^ coatings +; experimental culture): (**a**) confocal imaging showing nuclei organisation (blue—stained with DAPI), an actin cytoskeleton (green—Atto 488 Phalloidin), and the distribution of the mitochondrial network (red—mitoRed). The pictures were captured under magnification equal to 630× (scale bar = 40 μm); (**b**) the ultrastructure of the MC3T3-E1 cells. The pictures were captured using SEM under 500× magnification (scale bar = 100 μm and magnification equal to 2500×); (**c**) Dot-plots presenting distribution of cells based on mitochondrial membrane potential. Cells were separated into four populations: live (Live—bottom-right corner), live with the depolarised mitochondrial membrane (Depolarized/Live—bottom-left corner), dead with the depolarised mitochondrial membrane (Depolarised/Dead—upper-left corner), and dead (Dead—upper-right corner). The results of the statistical analysis showing the percentage of cells with (**d**) high mitochondrial potential and (**e**) total depolarised cells. (significant differences are marked with asterisks (* *p* < 0.05), non-significant results are marked as ns).

**Figure 9 materials-13-04817-f009:**
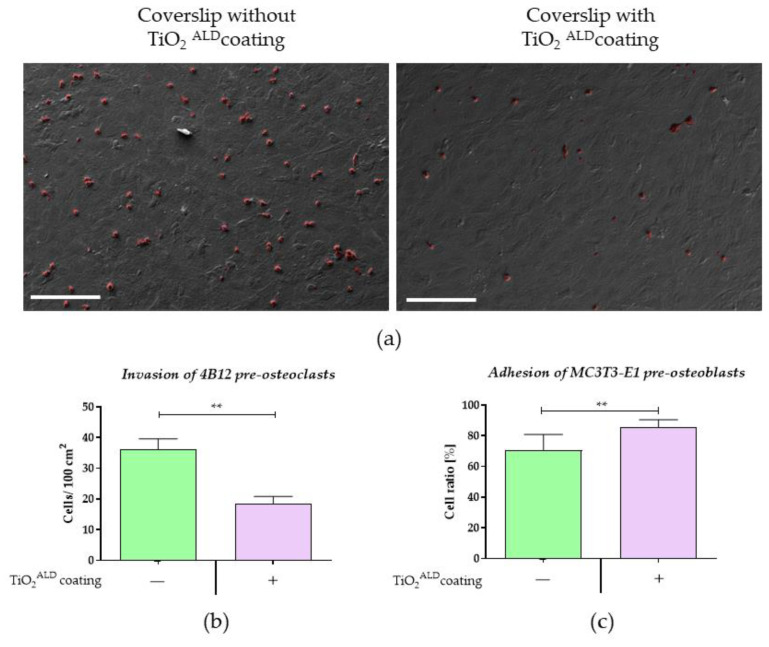
The invasion of preosteoclasts in the control co-culture (TiO_2_-) and the culture propagated on TiO_2_ coatings obtained by ALD (TiO_2_^ALD^ coatings+; experimental co-culture): (**a**) the representative pictures of MC3T3-E1 co-cultured with pre-osteoclasts. 4B12 cells were visualised and coloured red (GNU Image Manipulation Program 2.10.18). The pictures were captured using SEM under 500× magnification (scale bar = 20 µm); (**b**) the results of statistical analysis showing the number of preosteoclasts. (significant differences are marked with asterisks (** *p* < 0.01)).

**Figure 10 materials-13-04817-f010:**
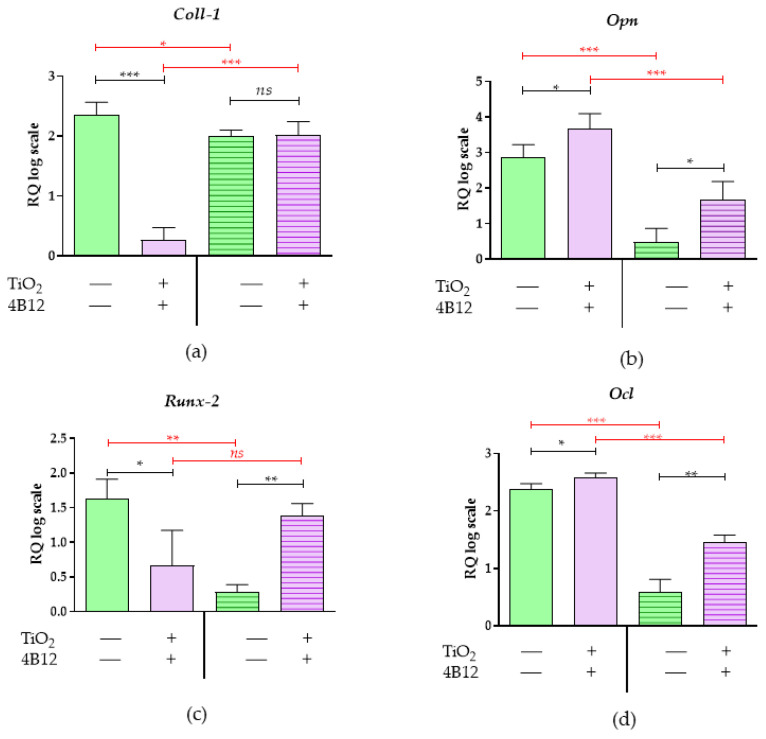
The mRNA expression of genes associated with osteogenic potential. The analysis of the control culture (TiO_2_
^ALD^ coatings -) and the culture propagated on TiO_2_coatings (TiO_2_
^ALD^ coatings +; experimental culture) examined (**a**) Coll-1, (**b**) Opn, (**c**) Runx-2, and (**d**) Ocl. The transcripts’ profiles were measured using the RT-qPCR technique. The relative quantification (RQ) was performed using the RQMAX method and the results are presented in a log scale. (significant differences are marked with asterisks (*** *p* < 0.001; ** *p* < 0.01; * *p* < 0.05), non-significant results are marked as ns).

**Figure 11 materials-13-04817-f011:**
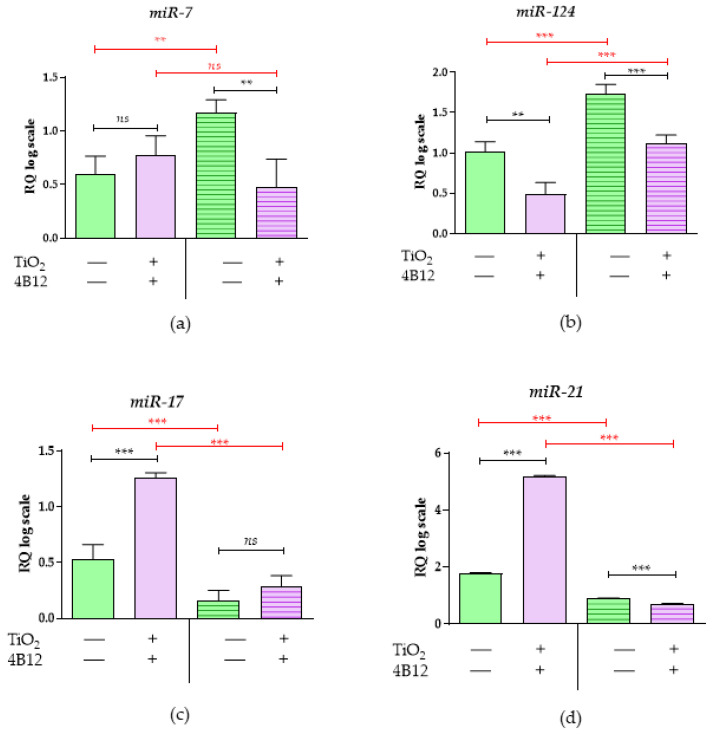
The expression of miRNA associated with osteogenic potential. The analysis of the control cultures (TiO_2_
^ALD^ coatings -) and the cultures propagated on TiO_2_ coatings obtained by ALD (TiO_2_
^ALD^ coatings +; experimental culture) examined (**a**) miR-7, (**b**) miR-124, (**c**) miR-17, and (**d**) miR-21. The transcripts’ profiles were measured using the RT-qPCR technique. The relative quantification (RQ) was performed using the RQ_MAX_ method, and the results are presented in a log scale. (significant differences are marked with asterisks (*** *p* < 0.001; ** *p* < 0.01), non-significant results are marked as ns).

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
