# Peer review of "Titanium Dioxide Thin Films Obtained by Atomic Layer Deposition Promotes Osteoblasts’ Viability and Differentiation Potential While Inhibiting Osteoclast Activity—Potential Application for Osteoporotic Bone Regeneration"

_materials, 2020, doi:10.3390/ma13214817_

Round 1

Reviewer 1 Report

Manuscript written by Smieszek et al. address interesting field of using nanoscale thin TiO2 films as substrates for bioapplications. 

-Written text, especially introduction and Materials and methods (but also in results for some extent) are quite superfluos in style and some condensation would make it better for the reader. 

-TiO2 prepared by other techniques has been used as an substrate for similar experiments, but these experiments are not properly referenced in the introduction.  At least some checking some review papers would make the introductions more useful. At least reviewing the main effects of surface morphology, porosity, composition etc. 

- Some of the most recent ALD TiO2 bio-related papers are missing from the Introduction, like DOI: 10.1039/D0RA05141A and https://doi.org/10.1021/acsabm.0c00871  Data reported in the manuscript should be compared to these.

-TiO2ALD is is not chemically correct formula and should not be used.

-How and how long the samples were stored before contact angle measurements. Was there any cleaning procedure before measurement?

-Only one type of ALD TiO2 is used, if would be informative that different crystallinities/morphologies were evaluated.  

-Carbon content in TiO2 is relatively high, which might also correlate to Ti:O ratio (in the form of carbonate species), this can be verified by FTIR for example

Reviewer 2 Report

The manuscript is very interesting. Congratulations!

I recommend few minor changes. There are few references missing in the introduction. The text of the pictures are too small. The pictures are nice, but the text should be better presented.

  • The duration of the pre-heating should be informed in this sentence "Additionally, the titanium precursor was preheated to 70°C."
  • - In the results: The authors do not compare their XPS results with previous published reports. Furthermore, they affirm that oxygen excess (ratio of O:Ti content was 2.4) was due to absorption of water and carbon oxide in the surface. This simple explanation is not enough to convince the readers.
  • - The authors confirm the formation of "amorphous titanium dioxide" without showing the XRD measurements. "We did not detect any additional signal in comparison to the broad peak of an uncoated cover glass (the data are not shown here)."
  • - References are missing not only in the introduction. Some parts in the discussion are also requiring references, for example, in these sentences "TiO2 occurs in various crystallographic phases—amorphous, rutile, anatase, and brookite—or it can coexist in several phases." / "The crystallinity affects the biofunctionality as well as the quality of the coating"
  • - The authors affirm that "Amorphous oxides obtained by low temperature are far from stoichiometric", but mention in the results "It confirms the formation of an amorphous titanium dioxide with stoichiometry close to the ideal. " These contradictions should be checked and corrected. Despite these minor revisions and improvement of data/graphics presentation that should be done, I'd like to mention that the study is very interesting and rich.

Reviewer 3 Report

The authors provide a paper dealing with the deposition of TiO2 thin films with biological applications. The paper can be of interest for Materials, but MAJOR revisions are requested.

  • The title is too long at it must be shortened and improved to get more the sense of the paper. Similarly, the abstract has to be improved providing only key concept for the paper.
  • In the introduction, I think it can be important to cite additional work dealing with nanostructured TiO2 films for biological applications such as doi.org/10.3762/bxiv.2020.29.v1
  • and doi.org/10.1016/j.matdes.2018.06.051. These paper and other should be included and the morphology of the TiO2 film must be discussed with respect to the PLD deposited ones.
  • Quality of the graphs (especially all the histograms) have to be improved. Up to now, they are of a very low quality and difficult to understand.
  • I would add a SEM images for films in cross-section. This would be good to observe the morphology of the film (with respect to the literature) and to comment of a potential effect of nanostructure, film thickness.
  • Addition of Raman analysis as in the previous Refs. will improve the paper.

Round 2

Reviewer 1 Report

The manuscript is clearly improved and it now can considered for the publication.

Reviewer 3 Report

-